# A Novel Long Non-Coding RNA-01488 Suppressed Metastasis and Tumorigenesis by Inducing miRNAs That Reduce Vimentin Expression and Ubiquitination of Cyclin E

**DOI:** 10.3390/cells9061504

**Published:** 2020-06-20

**Authors:** Syuan-Ling Lin, Yang-Hsiang Lin, Hsiang-Cheng Chi, Tzu-Kang Lin, Wei-Jan Chen, Chau-Ting Yeh, Kwang-Huei Lin

**Affiliations:** 1Translational Medicine Research Center, China Medical University Hospital, Taichung 404, Taiwan; irislin3316@gmail.com; 2Department of Biochemistry, School of Medicine, Chang-Gung University, Taoyuan 333, Taiwan; 3Liver Research Center, Chang Gung Memorial Hospital, Linko, Taoyuan 333, Taiwan; sam4915@yahoo.com.tw (Y.-H.L.); chauting@adm.cgmh.org.tw (C.-T.Y.); 4Radiation Biology Research Center, Institute for Radiological Research, Chang Gung University/Chang Gung Memorial Hospital, Linkou, Taoyuan 333, Taiwan; nonbalance@gmail.com; 5Neurosurgery, Fu Jen Catholic University Hospital and School of Medicine, Fu Jen Catholic University, New Taipei City 242, Taiwan; tklin100@cgmh.org.tw; 6Cardiovascular Division, Chang Gung Memorial Hospital, Chang Gung University College of Medicine, Taoyuan 333, Taiwan; wjchen@cgmh.org.tw; 7Research Center for Chinese Herbal Medicine, College of Human Ecology, Chang Gung University of Science and Technology, Taoyuan 333, Taiwan

**Keywords:** long intergenic non-coding RNAs, vimentin, cyclin E, microRNA-124-3p/138-5p, tumor suppressor, hepatocellular carcinoma

## Abstract

Long intergenic non-coding RNAs (lincRNAs) play important roles in human cancer development, including cell differentiation, apoptosis, and tumor progression. However, their underlying mechanisms of action are largely unknown at present. In this study, we focused on a novel suppressor lincRNA that has the potential to inhibit progression of human hepatocellular carcinoma (HCC). Our experiments disclosed long intergenic non-protein coding RNA 1488 (LINC01488) as a key negative regulator of HCC. Clinically, patients with high LINC01488 expression displayed greater survival rates and better prognosis. In vitro and in vivo functional assays showed that LINC01488 overexpression leads to significant suppression of cell proliferation and metastasis in HCC. Furthermore, LINC01488 bound to cyclin E to induce its ubiquitination and reduced expression of vimentin mediated by both miR-124-3p/miR-138-5p. Our results collectively indicate that LINC01488 acts as a tumor suppressor that inhibits metastasis and tumorigenesis in HCC via the miR-124-3p/miR-138-5p/vimentin axis. Furthermore, LINC01488 interacts with and degrades cyclin E, which contributes to its anti-tumorigenic activity. In view of these findings, we propose that enhancement of LINC01488 expression could be effective as a potential therapeutic strategy for HCC.

## 1. Introduction

Hepatocellular carcinoma (HCC) is one of the most common malignant tumor types and the third leading cause of cancer-related mortality worldwide [1,2]. The prognosis for HCC remains poor, with a one-year survival rate of ~40%, even after liver resection which is considered one of the most effective therapeutic strategies [3,4]. The majority of HCC patients present symptoms at advanced stages when metastasis has occurred. Metastasis not only interferes with the treatment options but also presents a significant cause of liver failure and tumor recurrence. Therefore, clarification of the mechanisms involved in HCC progression is essential to facilitate the development of effective novel therapeutic targets.

Long intergenic non-coding RNAs (lincRNAs) are a class of non-protein-coding RNAs more than 200 nucleotides in length [5]. LincRNAs are frequently dysregulated in various cancers and are shown to have multiple functions in several biological processes such as modulation of proliferation and metastasis and cellular development and differentiation [2,6,7]. Recent studies have revealed opposing roles of lincRNAs as either proto-oncogenes (TUG1, HOTAIR) or tumor suppressor genes (LINC01488, MEG3) in tumorigenesis [8,9,10].

Long intergenic non-protein-coding RNA 1488 (LINC01488) has been considered as a cyclin D long promoter-associated ncRNA (ncRNA_CCND1_) located within the promoter region of cyclin D, specifically binds an RNA-binding protein, Translocated in Liposarcoma (TLS), and exerts transcriptional repression through histone acetyltransferase (HAT) inhibitory activity [11]. Further, LINC01488 is located at chromosome 11q13.3 with a length of 3425 bp (from NCBI); however, LINC01488 directly located in the promoter region of cyclin D and bind with TLS will be verified. Another study suggests that LINC01488 and AP000439 cooperatively participate in the modulation of repair pathway choices in response to double-strand breaks in breast tumors [12]. However, the functional roles of LINC01488 in tumorigenesis and the underlying mechanisms remain unclear.

In this study, we investigated the potential involvement of LINC01488 in tumor metastasis and proliferation in HCC. Our data support novel inhibitory effects of LINC01488 on metastasis through the vimentin/miR-124-3p/miR-138-5p axis as well as proliferation through ubiquitination of cyclin E in HCC.

## 2. Materials and Methods

### 2.1. Cell Culture

The SK-Hep1 and Hep3B cell lines were cultured in Dulbecco’s modified Eagle’s medium (DMEM) containing 10% (*v*/*v*) fetal bovine serum (FBS). Cells were grown in a humidified atmosphere of 5% CO_2_ at 37 °C. The SK-Hep1 cell is a human hepatic adenocarcinoma cell line and considered a cell model for liver sinusoidal endothelial cells.

### 2.2. Quantitative Reverse Transcription-PCR (qRT-PCR)

To quantify lncRNA transcripts, total RNA was extracted from cells with TRIzol reagent (Life Technologies Inc., Carlsbad, CA, USA) and converted into cDNA using MMLV reverse transcriptase (Thermo Fisher Scientific Inc., Kalamazoo, MI, USA). The qRT-PCR was conducted in a 15 μL reaction mixture containing forward and reverse primers and 1X SYBR Green mix (Applied Biosystems, Carlsbad, CA, USA). Quantification of miRNA was performed as described previously [13]. Briefly, 10 μL reaction mixtures containing miRNA-specific stem-loop RT primers, dNTPs, MMLV reverse transcriptase, and 1 μg total RNA were used for the RT reaction. The qRT-PCR was conducted in a 10 μL reaction mixture containing miRNA-specific forward primer, universal reverse primer, and 1X SYBR Green mix (Applied Biosystems, SF, USA). The ABI QuantStudio 3 Real-Time PCR system (Thermo Fisher Scientific Inc.) was used for qRT-PCR analysis. The primer sequences are listed in Appendix A.

### 2.3. Immunoblot Analysis

The immunoblot procedure was performed as described previously [14]. Antibodies specific for cyclin E, CDK2, CDK4, fascin, MMP-9, vimentin, and Ub (Santa Cruz Biotechnology Inc., Santa Cruz, CA, USA), cyclin D, (Abcam, San Francisco, CA, USA), N-cadherin (Thermo Fisher Scientific Inc.), and GAPDH (Merck Millipore, Billerica, MA, USA) were employed. Band intensities were calculated using Image Gauge software (Fujifilm, Tokyo, Japan). The intensities of target gene expression were normalized to GAPDH signals for lysate and fascin signals for supernatant proteins.

### 2.4. Establishment of Gene Overexpression and Knockdown Stable Cell Lines

Construction of LINC01488-overexpressing plasmid was performed using the CRISPR system (Addgene, Watertown, MA, USA). We designed two guide RNA sequences of LINC01488 based on the MIT–CRISPR website, followed by cloning into the gRNA vector. Subsequently, hepatoma cells were co-transfected with the vector and dCAS-VP64_Blast plasmid. The gRNA cloning vector and dCAS-VP64_Blast plasmid were purchased from Addgene (Watertown, MA, USA). The specific shRNAs of LINC01488 were constructed into a shRNA cloning vector (Institute of Molecular Biology, Academia Sinica, Taiwan) using an annealing method. The following annealing conditions were used: 90 °C for 4 min, 70 °C for 10 min, 50 °C for 5 min, 37 °C for 30 min, and 25 °C for 10 min. The shRNA sense (tail with CCGG at the 5′end) and antisense (tail with AATT at the 5′end) oligonucleotides were designed and anneal mixture containing sense, antisense oligonucleotides, and 10× annealing buffer by PCR system. Single shRNA plasmid and virus package plasmids (pCMV-∆R8.91 and pMD.G) were co-transfected into 293FT cells and the virus harvested after 24 h of transfection. Oligonucleotides encoding pre-miR-124-3p and -138-5p sequences were cloned into the pcDNA6.2GW/EmGFP-miR expression vector (Invitrogen, Kalamazoo, MI, USA) to facilitate pre-miRNA insertion within the 3′untranslated (3′-UTR) region of the green fluorescent protein (GFP) gene. LacZ, not predicted to target any known vertebrate genes, was used as the control. The miR-124-3p and miR-138-5p inhibitors were purchased from Dharmacon (Lafayette, CO, USA).

### 2.5. Luciferase Reporter Assay

The partial 3′-UTR of human vimentin was amplified from cDNA via PCR and cloned into the SpeI/MluI sites of pMIR-REPORT firefly luciferase vector (Applied Biosystems). The putative miR-124-3p and miR-138-5p binding sites within vimentin 3′-UTR were amplified via PCR and validated by sequencing. The miR-124-3p- and miR-138-5p-overexpressing and control cells were seeded and co-transfected with reporter plasmids and the pSVβ vector (Clontech Laboratories, Inc., Mountain View, CA, USA) using TurboFectTM (Fermentas, Glen Burnie, MD, USA). After transfection for 24 h, luciferase activity was measured and compared with that of the control group. Luciferase activity was normalized to that of β-galactosidase.

### 2.6. In Vitro Migration and Invasion Assays

Cell migratory and invasive capabilities were determined using a rapid in vitro Transwell assay (Becton–Dickinson, Franklin Lakes, San Jose, NJ, USA) with a membrane pore size of 8 μm. Cell density was adjusted to a concentration of 3 × 10^4^ cells/mL, and 100 μL suspension seeded on either non-Matrigel-coated (migration assay) or Matrigel-coated (invasion assay) (Becton-Dickinson) upper chambers of the Transwell plate. For both assays, the medium in the upper chamber was serum-free DMEM and that in the lower chamber was DMEM supplemented with 20% FBS. After incubation for 20 h at 37 °C, cells traversing the filter from the upper to lower chamber were examined via crystal violet staining. Migratory or invasive cells were counted using ImageJ software.

### 2.7. In Vitro Proliferation Assay

The tumor cell growth rate was determined using the cell proliferation assay. Cell density was adjusted to 1 × 10^5^ cells for seeding in wells of a 6 well plate. Viable cells were trypsinized and counted on the indicated days (days 2 to 7).

### 2.8. Flow Cytometry

Cells were harvested via trypsin digestion and fixed in 70% ethanol overnight at 4 °C. Next, cells were treated with 0.5% Triton X-100 and 0.05% RNase A for 1 h at 37 °C. Thereafter, nuclear DNA was incubated with 50 μg/mL propidium iodide for 30 min at 4 °C and cells analyzed on a FACSCalibur flow cytometer (Becton–Dickinson Immunocytometry Systems, CA, USA).

### 2.9. RNA Immunoprecipitation (RIP) Assay

The RIP assay was performed as described previously [15]. Antibodies against cyclin E and vimentin for this experiment were purchased from Santa Cruz.

### 2.10. Ubiquitination Assay

Control and LINC01488-overexpressing cells were treated with MG132 (20 μM) for an additional 4 h and extracted using lysis buffer containing protease inhibitors. Cell lysates were incubated with protein A/G (GE Healthcare) for 1 h to prevent non-specific binding. Next, products were incubated overnight at 4 °C with cyclin E antibody (Santa Cruz Biotechnology Inc.) and precipitated with protein A/G (GE Healthcare) for 1 h at 4 °C. The ubiquitinated cyclin E signal was detected with the Ub antibody (Santa Cruz Biotechnology Inc.).

### 2.11. Animal Models

In model I, nude mice were subcutaneously injected with LINC01488-depleted or -overexpressing SK-Hep1 cells (1 × 10^6^) to assess the effects on tumor formation ability. Tumor volumes (mm^3^) were measured using the formula (W^2^ × L)/2 (W, smallest diameter; L, longest diameter). In model II, severe combined immunodeficient (SCID) mice were employed to determine the invasive capability of LINC01488-depleted and -overexpressing SK-Hep1 cells following intravenous injection (2 × 10^6^ cells). All animals were sacrificed on week 8 after tumor inoculation, and livers and lungs removed. Formaldehyde-fixed and paraffin-embedded tissues from lungs of SCID mice were examined by immunohistochemistry (IHC) assay using vimentin, cyclin E (Santa Cruz) and cyclin D (Abcam) antibody. Positive staining, indicating tumor cells, appeared as a brown color showing vimentin, cyclin E and cyclin D immunoreactivity. Animal experiments were performed according to the guidelines of United States National Institutes of Health and the Chang Gang Institutional Animal Care and Use Committee Guide for the Care and Use of Laboratory Animals.

### 2.12. Statistical Analysis

Results are presented as the means ± SD of three independent experiments. Statistical analysis was performed with SPSS version 15 software (SPSS Inc., Chicago, IL, USA) using the Mann–Whitney test for comparison of two groups and one-way ANOVA followed by Tukey’s post-hoc test for two or more groups. Kaplan–Meier curves were employed to analyze survival outcomes. Overall survival (OS) with death as an event was analyzed using the log-rank test. The *p*-values < 0.05 were considered significant.

## 3. Results

### 3.1. LINC01488 Downregulation in Human HCC Tissues Was Associated with Favorable Prognosis

Several differentially expressed lincRNAs were identified in the two HCC samples compared to adjacent normal tissues in our microarray analysis. Top 10 dysregulated lincRNAs are shown in Appendix A. Among these, 279 were consistently downregulated and 123 upregulated in HCC (fold change ≥ 2 or ≤ 0.5; *p* < 0.05). Notably, the role of suppressor lincRNAs in cancer progression is largely unknown. Hence, the current goal is to study the potential function or mechanism of tumor suppressor lincRNAs. LINC01488 displaying low expression in HCC samples was confirmed via qRT-PCR (Figure 1A). Furthermore, LINC01488 displayed significant negative correlation with TNM stage, tumor size, and pathological stage (Figure 1B–D). Importantly, HCC patients displaying low-LINC01488 expression showed dramatically poorer overall and recurrence-free survival rates (Figure 1E,F). According to the median ratio of relative LINC01488 expression, HCC patients were divided into high (LINC01488 expression ratio ≥ median ratio) and low (LINC01488 expression ratio < median ratio) groups. In terms of clinically significant findings, the higher expression group was negatively associated with gender (*p* = 0.007), tumor size (*p* = 0.009), stage (*p* = 0.045), and pathological stage (*p* = 0.05) (Table 1). Our data clearly show that LINC01488 was downregulated in HCC and higher expression was associated with favorable prognosis. Thus, LINC01488 was selected for further study due to the fact of its correlation with clinical significance.

### 3.2. Cell Proliferative Ability Is Enhanced upon Knockdown of LINC01488 In Vitro

Depletion of LINC01488 was established in SK-Hep1 and Hep3B cells via the lentivirus-based system (Figure 2A) and stable SK-Hep1 and Hep3B cell lines overexpressing LINC01488 generated using the gRNA–CRISPR system (Figure 2B). Previous studies have shown that LINC01488 is located within the promoter region of Cyclin D1 [11]. Furthermore, LINC01488 is reported to bind translocated in liposarcoma (TLS) and induce transcriptional repression through HAT inhibitory activity in addition to negatively regulating Cyclin D1 transcription to inhibit cell growth. Accordingly, we validated whether LINC01488 could suppress cell proliferation in vitro. Our experiments showed increased proliferation of SK-Hep1 and Hep3B cells after knockdown of LINC01488 (Figure 2C). Conversely, proliferation was significantly inhibited in LINC01488-overexpressing SK-Hep1 and Hep3B cells, compared with the control group (Figure 2D). The cell cycle was additionally analyzed via flow cytometry in LINC01488 knockdown or overexpressing Hep3B stable cells. By flow cytometry assay, the percentage of cell numbers were increased in LINC01488-depleted stable cells (34.7% or 40.2%) compared with the siRNA control cells (23.8% or 26.1%) in Hep3B at the S phase. The result indicated that knockdown expression of LINC01488 promoted cell cycle arrest at the S phase (Figure 2E). Conversely, LINC01488-overexpressing cells were arrested to a lower extent at the S phase, compared to the gRNA vector-transfected cells (27.1% versus 34.9%) (Figure 2F). Taken together, our results indicate that LINC01488 significantly inhibits proliferation of liver cancer cells.

### 3.3. Upregulation of LINC01488 Suppresses Hepatoma Cell Migration and Invasion Ability In Vitro

To examine whether LINC01488 affects cell mobility in liver cancer, in vitro migration and invasion assays were performed. Knockdown of LINC01488 dramatically promoted the migratory and invasive abilities of SK-Hep1 and Hep3B cells, compared with the cell lines expressing control shRNA (Figure 3A–D). Conversely, mobility was distinctly suppressed in LINC01488-overexpressing cell lines (Figure 3E,F). The finding that LINC01488 dramatically reduced migratory/invasive capacities supports a role as a tumor suppressor in hepatoma cells.

### 3.4. EMT Markers Were Negatively Regulated by LINC01488

Our experimental results clearly suggest that upregulation of LINC01488 leads to suppressed tumor cell mobility. Moreover, mRNA and protein levels of vimentin, N-cadherin, and MMP-9 were significantly increased in LINC01488 knockdown hepatoma cells, compared with cell lines transfected with control shRNA (Figure 4A,B). Similarly, the fluorescence signals of vimentin and N-cadherin were stronger in LINC01488 knockdown hepatoma cells than in control shRNA HCC cells by immunofluorescence analysis (Appendix A). Conversely, stable SK-Hep1 and Hep3B cell lines overexpressing LINC01488 showed reduced vimentin, N-cadherin, and MMP-9 mRNA and protein levels relative to the control group (Figure 4C,D). Likewise, the fluorescence signals of vimentin and N-cadherin were declined in LINC01488-overexpressing hepatoma cells (Appendix A). GAPDH and fascin were used as the loading controls for lysate and supernatant samples, respectively. In addition, to determine whether the vimentin is the major molecular target of anti-metastasis or anti-proliferation, the knockdown of vimentin was performed. We observed the knockdown of vimentin in LINC01488-depleted stable cell lines repressed the migratory/invasive capability of knockdown LINC01488-promoted metastasis. However, knockdown of vimentin didn’t affect cell proliferative ability (Appendix A). Based on the results, we suggest that LINC01488 negatively regulates the epithelial–mesenchymal transition (EMT) in liver cancer cells.

### 3.5. LINC01488 Modulates miR-124-3p/miR-138-5p to Suppress Metastasis through Targeting Vimentin

Next, we investigated the mechanisms by which LINC01488 negatively regulates metastasis in liver cancer cells. Recent research suggests that lincRNAs modulate a variety of biological processes through binding proteins and altering their functions [16,17]. Vimentin is an EMT-related protein negatively regulated by LINC01488, as determined with the RNA immunoprecipitation (RIP) assay. Accordingly, we examined whether LINC01488 directly interacts with vimentin in SK-Hep1 and Hep3B cells, which revealed no interactions between the two molecules (Appendix A). The Target Scan Human7.1 website was used to identify the miR-124-3p and miR-138-5p potentially binding to vimentin. To further validate whether lincRNA suppresses motility mediated by miRNAs, we determined the expression patterns of the above two miRNAs in response to altered expression of LINC01488. Notably, miR-124-3p and miR-138-5p levels were dramatically increased in SK-Hep1 and Hep3B-overexpressing LINC01488 relative to the gRNA vector-transfected control line, as determined via q-PCR (Figure 5A). We had identified putative miR-124-3p- and miR-138-5p binding sites in the 3′-untranslated region (3′-UTR) of vimentin mRNA using Target Scan Human 7.1 (Figure 5B, top). Therefore, we subsequently established a reporter vector consisting of the luciferase coding sequence followed by 3′-UTR of vimentin. The effects of overexpression of miR124-3p and miR138-5p on vimentin reporter activity were further analyzed. Overexpression of miR-124-3p and miR-138-5p led to suppression of the luciferase activity of VIM 3′-UTR by 50–60% in SK-Hep1 and Hep3B cells, compared with the pcDNA6.2 vector-transfected control cell line (Figure 5B, middle). Conversely, luciferase activity was increased in the inhibitor of miR-124-3p or miR138-5p in SK-Hep1 and Hep3B cells (Figure 5B, bottom). To further determine whether miR-124-3p and miR-138-5p are involved in LINC01488-mediated inhibitory effects on metastasis, miR124-3p or miR135-5p-containing or control plasmids were transfected into SK-Hep1- or Hep3B-control or linc01488-depleted stable cell lines. Migratory capacity was decreased upon overexpression of miR-124-3p or miR-138-5p in stable SK-Hep-1 and Hep3B cells with LINC01488 knockdown, compared to control cells (Figure 5C). The quantification of invasiveness data was presented in Figure 5C (bottom). We observed increased vimentin protein expression following knockdown of LINC01488. However, the vimentin level was decreased under conditions of overexpression of miR-124-3p or miR-138-5p in LINC01488-depleted stable cells (Figure 5D). Our results collectively suggest that LINC01488 upregulates miR-124-3p and miR-138-5p expression to suppress vimentin expression.

### 3.6. LINC01488 Binds Cyclin E and Decreases Its Protein Levels via Ubiquitination

To determine the mechanisms underlying LINC01488 suppressor activity in HCC, protein levels of cyclin E, cyclin D, CDK2, and CDK4 were evaluated. Levels of these proliferation-related proteins were significantly increased by knockdown of LINC01488 in SK-Hep1 and Hep3B stable cell lines, compared with those in control cells (Figure 6A) while overexpression of LINC01488 was associated with decreased expression (Figure 6B). A previous study has shown that transcriptional repression of cyclin D is decreased by LINC01488 binding to TLS [11]. Here, we additionally examined cyclin E, which is a well-known proliferative marker. To determine whether the cyclin E is a major molecular for functions of pro-metastasis or pro-proliferation, the knockdown experiment was performed. The result showed that the cyclin E depleted-cells significantly reduced the proliferative capability in LINC01488-depleted cells. On the other hand, knockdown of cyclin E did not influence cell metastasis (Appendix A). The interactions between LINC01488 and cyclin E have not been reported to date. Interestingly, expression of this protein was negatively regulated by LINC01488. To further analyze the potential interactions between LINC01488 and cyclin E, the RIP assay was performed using an anti-cyclin E antibody. We observed significantly higher enrichment of LINC01488 with the anti-cyclin E antibody, compared with the non-specific IgG control (Figure 6C), indicative of specific associations between LINC01488 and cyclin E. Further, CDKs family, such as CDK2 and CDK4 did not interact with LINC01488 by RIP assay (Appendix A). Importantly, no marked changes in cyclin E mRNA levels were observed (Appendix A). Accordingly, we hypothesize that LINC01488 binds to cyclin E and exerts effects at the translational level. To validate this theory, LINC01488-overexpressing SK-Hep1 or Hep3B cells were treated with cycloheximide (CHX) or the proteasome inhibitor MG-132. The cyclin E protein level was decreased by CHX in LINC01488-overexpressing SK-Hep1 or Hep3B, compared with gRNA-vector cell lines (Figure 6D, top). Similarly, MG-132 stabilized degradation of cyclin E protein in LINC01488-overexpressing hepatoma cells (Figure 6D, bottom). The immunoprecipitation (IP) assay was employed to ascertain whether polyubiquitin-attached cyclin E protein is increased in MG-132-treated LINC01488-overexpressing Hep3B stable cells, compared with MG-132-treated gRNA vector-transfected cells (Figure 6E, lane 3 versus 4). The data suggest that cyclin E degradation occurs via effects of LINC01488 on the ubiquitin-proteasome pathway.

### 3.7. LINC01488 Suppresses Metastasis and Tumorigenesis of Hepatoma In Vivo

To evaluate the influence of LINC01488 on lung metastasis in vivo, a xenograft mouse model was used. LINC01488-overexpressing cells (2 × 10^6^) and SK-Hep1 stable cells transfected with sh_LINC01488 or appropriate controls (gRNA vector) (Figure 7A) were injected into the tail vein of SCID mice. Data were assessed after eight weeks of injection using the immunohistochemistry (IHC) assay. Knockdown of LINC01488 led to dramatic enhancement of hepatic metastasis (observed via hematoxylin and eosin (H&E) staining) and increased protein levels of vimentin (based on IHC staining). Conversely, the vimentin protein level was reduced in LINC01488-overexpressing SK-Hep1 stable cells (Figure 7B). To investigate the effects of LINC01488 on proliferation rates in vivo, control, sh_LINC01488, gRNA-vector or LINC01488-overexpressing SK-Hep1 stable cells (1 × 10^6^) were subcutaneously administered to nude mice. We observed that upon knockdown of LINC01488, xenografts grew faster and were larger in size than control xenograft tumors. As expected, tumor volumes of LINC01488-overexpressing xenografts were smaller than those of gRNA vector-containing xenografts (Figure 7C). Notably, xenograft tumors with knockdown of LINC01488 displayed increased levels of cyclin E and cyclin D proteins, supporting faster growth than control tumors. On the other hand, cyclin E and cyclin D levels were decreased in xenograft tumors overexpressing LINC01488 (Figure 7D,E). Our collective data validate the inhibitory effects of LINC01488 on HCC development and metastasis.

## 4. Discussion

In view of the continued poor prognosis of HCC patients worldwide, early detection and surgical therapy are essential for improving survival rates. LincRNAs are reported to have significant therapeutic potential in several human cancer types including HCC [18]. Recent studies have shown that dysregulation of lincRNAs leads to altered biological roles in HCC. For example, Huang et al. [19] suggested that lncRNA–Dreh overexpression suppresses tumor metastasis in vivo in an orthotopic liver implanted metastasis model through decreasing vimentin expression. Yuan and co-workers [7] demonstrated that lncRNA–ATB induces EMT and promotes the invasion-metastasis cascade of HCC cells in vivo via competitive binds to the miR-200 family to induce ZEB1/2 expression. Another study by Wang et al. [20] showed that linRNA–hPVT1 promotes cell proliferation, cycling and stem cell-like phenotype of hepatoma cells by enhancing NOP2 protein stability in vitro. Moreover, a number of lincRNAs have been shown to play important roles in HCC by our group. Taurine upregulated gene 1 (TUG1) regulates glycolysis and metastasis through miR-455-3p, AMPKβ2 and HK2 in HCC cells [2] and suppression of BC200 negatively modulates tumorigenesis by the thyroid hormone in HCC [21]. However, limited reports have recently addressed the biological activities and their underlying mechanisms of lincRNAs in HCC. Data from the current study disclosed significant downregulation of a novel lincRNA, LINC01488, in HCC. Notably, LINC01488 expression in HCC tissues was negatively associated with tumor size, TNM stage and pathological stage and comparison of Kaplan-Meier survival curves suggested poorer prognosis of HCC patients with lower LINC01488 expression. Our collective results support a tumor suppressor role of LINC01488 in HCC. Loss- and gain-of-function approaches consistently showed that LINC01488 markedly suppresses HCC cell proliferation and metastasis, both in vitro and in vivo. We further focused on the mechanisms underlying LINC01488-mediated suppression of HCC metastasis. The levels of EMT markers examined, including vimentin, N-cadherin and MMP-9, were decreased under conditions of overexpression of LINC01488 in HCC cells. Importantly, recent studies have shown that lincRNAs can function as miRNA sponges, decreasing the amount of miRNAs available to target mRNAs [22,23]. Previous studies have demonstrated that lincRNAs play important roles as regulators of metastasis through binding microRNAs (miRNAs) in multiple cancers. For instance, HOTAIR is a miR-148a sponge that regulates Slug to promote EMT expression in esophageal cancer [24], lincRNA-ROR induces EMT through ZEB regulation by inhibition of miR-205 in breast cancer [25] and H19 operates as a sponge for let-7 to increase HMGA-2 mediated EMT in pancreatic cancer [26]. Results obtained using the TargetScanHuman7.1 website showed that 3′-UTR of vimentin is targeted by miR124-3p and miR138-5p, which was confirmed via the vimentin 3′-UTR assay. The luciferase activity of vimentin 3′-UTR was significantly decreased upon overexpression of miR-124-3p and miR-138-5p. MiR-124-3p and miR-138-5p are reported to inhibit metastasis in various cancer types through distinct signaling pathways. For instance, miR-138-5p inhibits metastasis and EMT by targeting vimentin in breast cancer and renal cell carcinoma [27,28], and a novel feedback loop exists between miR-124-3p and the TGF-β pathway driving NSCLC metastasis [29]. However, our data showed that that LINC01488 positively regulates miR-124-3p and miR-138-5p to reduce vimentin expression in hepatoma cells, indicating that lincRNAs not only act as miRNA sponges but also regulators to modify target mRNA levels. The underlying mechanisms, including the pathways by which LINC01488 affects miRNA biogenesis to repress target gene expression, require further investigation [30,31]. For example, other lincRNAs affect miRNA biogenesis, linc-MD1 generates miR-206 and miR-133b to control muscle differentiation, lincRNAs could interact with miRNAs to regulate miRNA-mRNA interactions. Further, lncRNA-Uc.283+A prevents pri-miR-195 cleavage by Drosha to control pri-miR-195 processing [31,32,33]. In further, we’ll investigate whether the LINC01488 regulates microRNA biogenesis through promoting pri-miRNAs cleavage by Drosha. Further, Previous studies indicate that DNA-bound form of cyclin D1 can regulate gene expression, including miRNAs, through association of transcription factor and histone modulatory genes. Yu et al. group demonstrated that miR-17-5p and miR-20 were upregulated by cyclin D1 [34]. Meanwhile, cyclin D1 was a potential target gene of miR-17-5p and miR-20. Thus, a novel cyclin D1/miR-17-5p/miR-20 forms a regulatory feedback loop in breast cancer. Recently, differentially expressed miRNAs in a breast cancer cell line were determined using high-throughput screening [35]. These findings suggest that cyclin D1 acts as a regulator of miRNA. In the current study, we found that cyclin D1 was regulated by linc01488. Notably, precursor form of miR-124-3p and miR-138-5p transcripts were induced upon linc01488 overexpression. Based on those observations, we suggest that miR-124-3p and miR-138-5p probably were regulated by linc01488 through modulation of cyclin D1.

The protein levels of cyclin E, cyclin D, CDK2, and CDK4 were dramatically induced upon knockdown of LINC01488 in hepatoma cells. Multiple studies suggest that cyclin E plays an important role in promoting S phase entry and enhances cell cycle progression [36,37]. In our cell cycle analysis, overexpression of LINC01488 inhibited G1 progression and suppressed S phase entry through effects on cyclin E function. Interestingly, the cyclin E mRNA level was not altered by LINC01488 regulation, indicating specific effects of LINC01488 on cyclin E at the translation level in our experiments. Results from the RIP assay suggest that LINC01488 binds to cyclin E and decreases expression of the protein via the ubiquitin-proteasome pathway.

In conclusion, these results pointed out LINC01488 acts as a tumor suppressor in HCC. Mechanistically, LINC01488 positively affects miR-124-3p and miR-138-5p biogenesis to reduce vimentin expression and negatively regulates cyclin E via ubiquitination, leading to inhibition of metastasis and tumorigenesis in HCC. Therefore, enhancement of LINC01488 expression could represent a potential therapeutic strategy for HCC.

## Figures and Tables

**Figure 1 cells-09-01504-f001:**
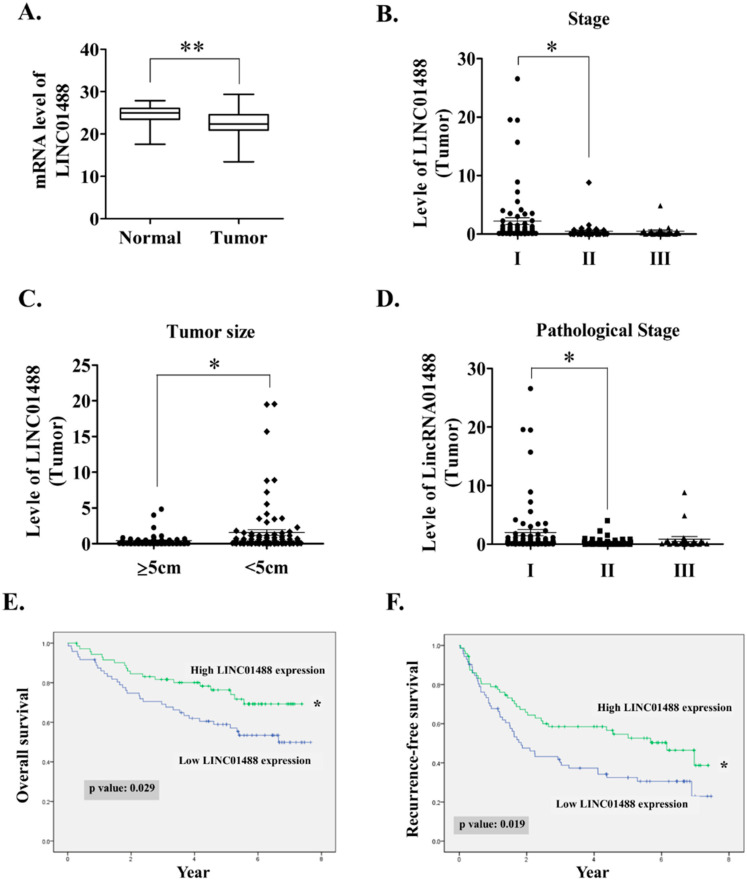
LINC01488 downregulation in human HCC tissues associated with favorable prognosis. (**A**) LINC01488 mRNA is highly expressed in normal relative to tumor tissue in 144 paired HCC specimens. Total RNA was isolated and analyzed via qRT-PCR. (**B**–**D**) Analysis of the clinicopathological significance of LINC01488 in HCC tumor (N:144) (**E**) Overall survival and (**F**) recurrence-free survival based on LINC01488 expression in HCC specimens determined using Kaplan–Meier analysis (N:144). Median expression levels of LINC01488 were used as the cut-off. Data are presented as means ± SD (* *p* < 0.05; ** *p* < 0.01).

**Figure 2 cells-09-01504-f002:**
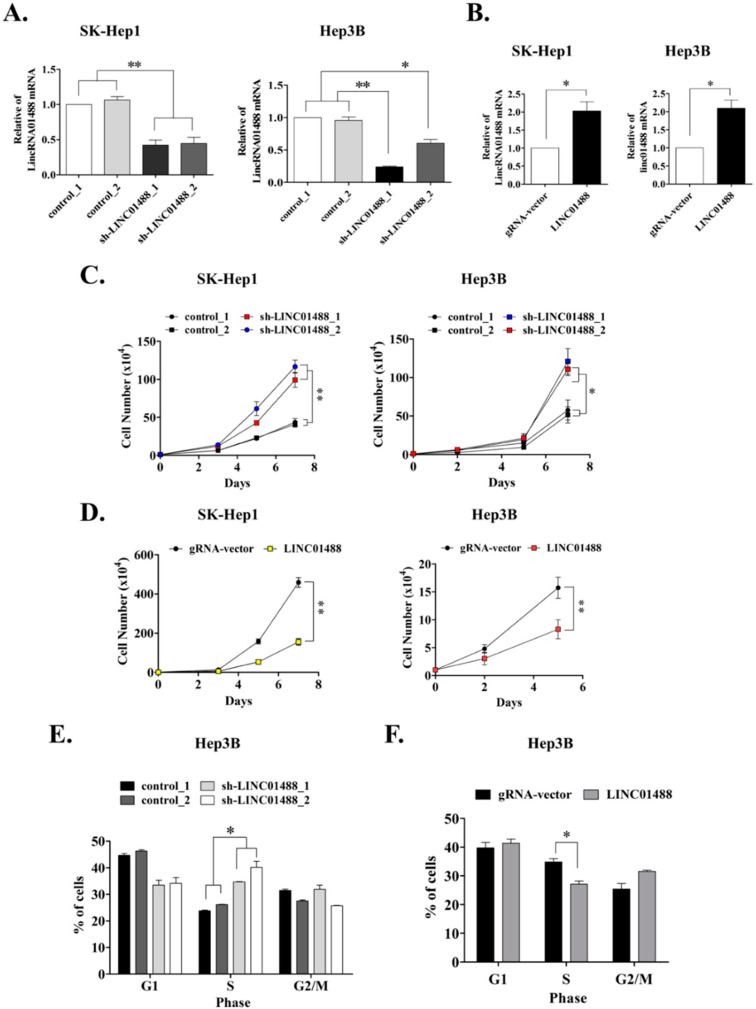
Proliferative ability was enhanced upon knockdown of LINC01488 in vitro. (**A**) Knockdown and (**B**) overexpression of LINC01488 in SK-Hep1 or Hep3B cell lines followed by q-RT-PCR analysis. (**C**) Knockdown of LINC01488 promotes proliferation of SK-Hep1 and Hep3B cells. (**D**) Overexpression of LINC01488 inhibits cell growth. (**E**) FACS analysis of the cell cycle. Depletion of LINC01488 promoted Hep3B cell arrest in the S phase. (**F**) Overexpression of LINC01488 in Hep3B inhibited the S phase of cells. Data are presented as means ± SD (* *p* < 0.05; ** *p* < 0.01).

**Figure 3 cells-09-01504-f003:**
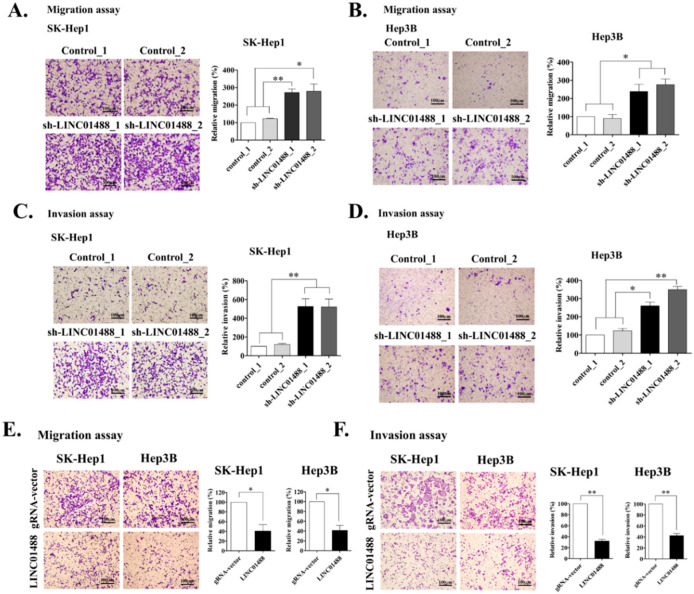
Upregulation of LINC01488 suppresses hepatoma cell migration and invasion ability in vitro. Migration assay on LINC01488 knockdown (**A**) SK-Hep1- and (**B**) Hep3B cells. Invasion assay on (**C**) SK-Hep1- and (**D**) Hep3B cells depleted of LINC01488. (**E**) Migratory and (**F**) invasive abilities of LINC01488-overexpressing SK-Hep1 or Hep3B cells in vitro. Quantified data are shown next to each graph. Data are presented as means ± SD (* *p* < 0.05; ** *p* < 0.01).

**Figure 4 cells-09-01504-f004:**
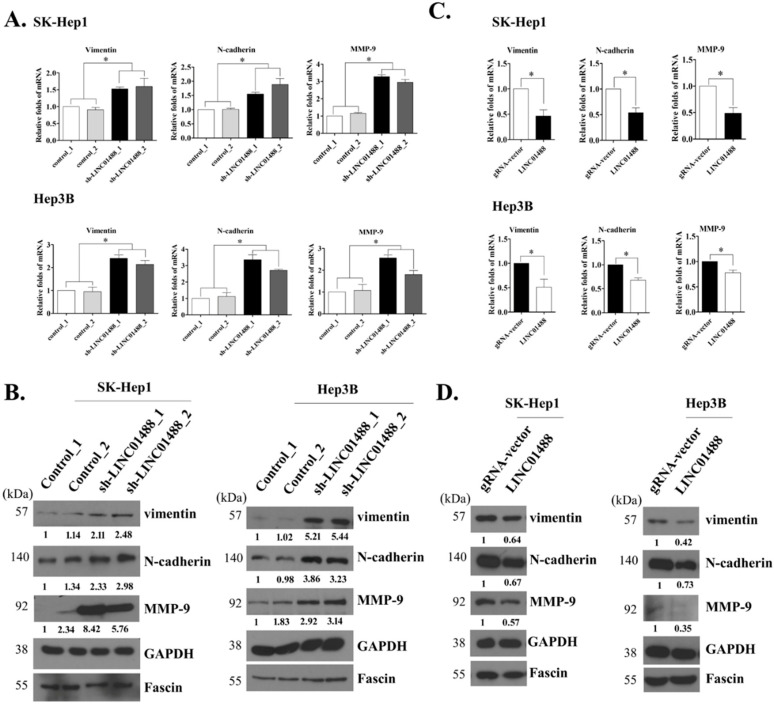
EMT markers negatively regulated by LINC01488. Depletion of LINC01488 affected EMT expression levels in SK-Hep1 or Hep3B cells as determined via (**A**) q-RT-PCR or (**B**) Western blot analysis. Upregulation of LINC01488 suppressed EMT expression levels in SK-Hep1 or Hep3B cells as determined via (**C**) q-RT-PCR or (**D**) Western blot analysis. GAPDH and fascin were used as the loading controls. Data are presented as means ± SD (* *p* < 0.05; ** *p* < 0.01).

**Figure 5 cells-09-01504-f005:**
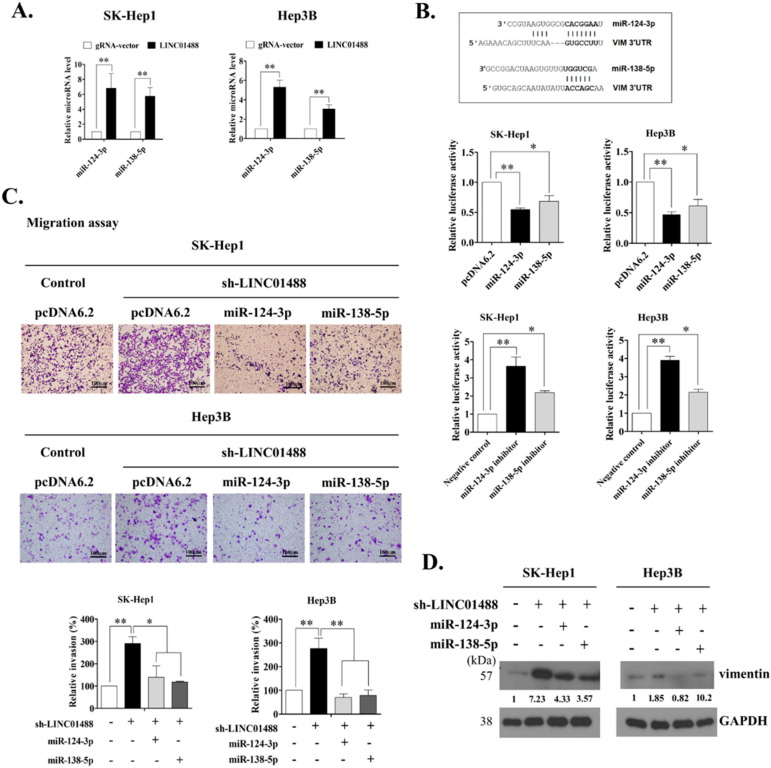
LINC01488 modulates miR-124-3p/miR-138-5p levels to suppress metastasis via targeting vimentin. (**A**) Increased miR-124-3p and miR-138-5p mRNA levels in LINC01488-overexpressing SK-Hep1 and Hep3B cells analyzed via q-RT-PCR. (**B**) MiR-124-3p and miR-138-5p target sequences on vimentin (top panel). Control (pcDNA6.2), miR-124-3p- or miR-138-5p-overexpressing SK-Hep1 or Hep3B cells were transfected with vimentin 3′-UTR, and luciferase activity calculated as fold change relative to control cells (middle panel). Negative control, miR-124-3p inhibitor or miR-138-5p inhibitor were additionally co-transfected with vimentin 3′-UTR in SK-Hep1 and Hep3B cells, and luciferase activity calculated as fold change relative to negative control cells (bottom panel). (**C**) Effects of miR-124-3p- or miR138-5p overexpression in LINC01488-depleted SK-Hep1 or Hep3B cells on cell migration and invasion assayed using the Transwell method. (**D**) Western blot analysis of vimentin protein levels in miR-124-3p-, miR-138-5p or LINC01488-depleted SK-Hep1 or Hep3B cells using GAPDH as the loading control. Data are presented as means ± SD (* *p* < 0.05; ** *p* < 0.01).

**Figure 6 cells-09-01504-f006:**
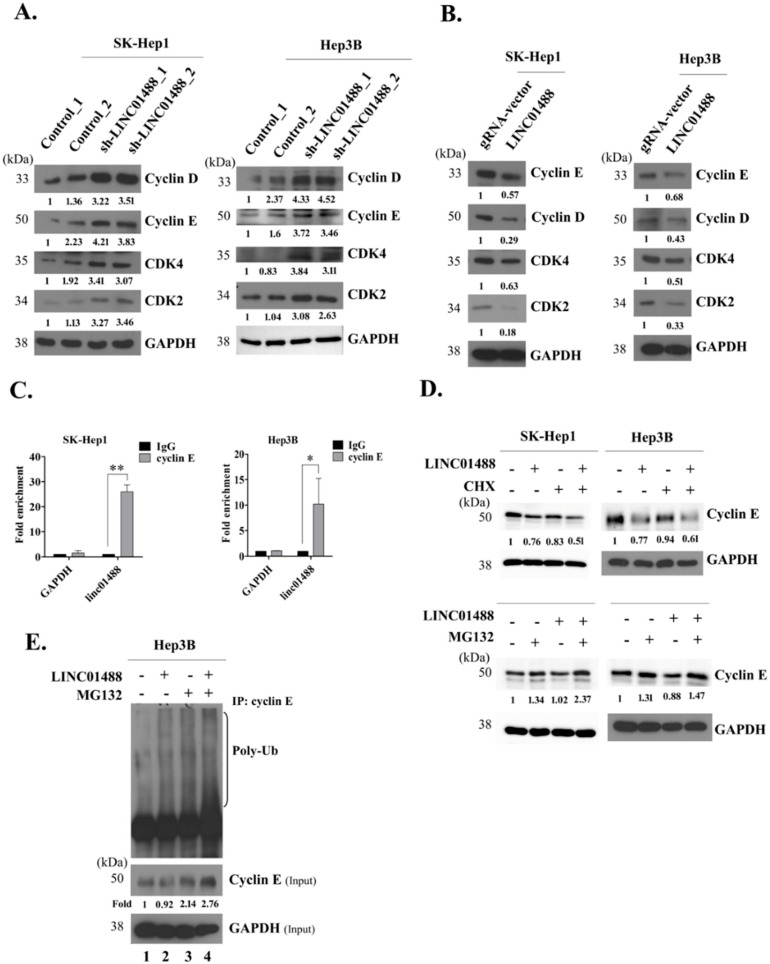
LINC01488 binds cyclin E and suppresses its protein levels via ubiquitination. Western blot analysis of cyclin E, cyclin D, CDK4, and CDK2 protein levels in LINC01488 (**A**) depleted and (**B**) overexpressed SK-Hep1 or Hep3B cells. (**C**) RNA immunoprecipitation assay to determine interactions between LINC01488 and cyclin E. (**D**) Effects of CHX (10 μM) and MG132 (20 μM) treatment on stability of cyclin E protein in LINC01488-overexpressing cells evaluated via Western blot. (**E**) Effects of MG132 (20 μM) and LINC01488 overexpression on cyclin E protein ubiquitination detected with an anti-Ub antibody. Data are presented as means ± SD (* *p* < 0.05; ** *p* < 0.01).

**Figure 7 cells-09-01504-f007:**
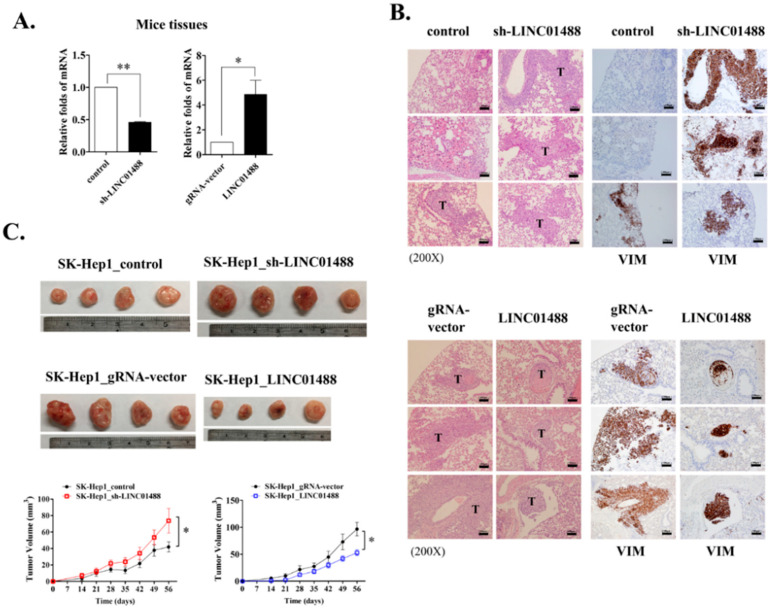
LINC01488 suppresses metastasis and tumorigenesis of hepatoma cells in vivo. Mice were injected with control, LINC01488-depleted or LINC01488-overexpressing SK-Hep1 or Hep3B cells either subcutaneously or via the tail vein. (**A**) qRT-PCR analysis of LINC01488 mRNA levels in lung tissue of mice. (**B**) Immunohistochemical analysis of vimentin protein levels in lung tissue of mice containing LINC01488-depleted or -overexpressing cells. (*n* = 3/group). (**C**) Tumor sizes from LINC01488-depleted or -overexpressing cells. Mice were subcutaneously injected with tumor cells for 8 weeks. (**D**) Immunohistochemical analysis of cyclin E and (**E**) cyclin D protein levels in lung tissue of mice injected subcutaneously with the indicated cells. (*n* = 4/group). Data are presented as means ± SD (* *p* < 0.05; ** *p* < 0.01). Scale bar: 100 μm.

**Table 1 cells-09-01504-t001:** Characterization of LINC01488 expression in HCC patients determined by qRT-PCR.

Clinicopathological Parameters		Number of Cases (*n* = 144)	LINC01488 Expression		*p*-Value
			Low (*n* = 73)	High (*n* = 71)	
Gender					0.0077 **
	Male	79	48 (65.7%)	31 (43.7%)	
	Female	65	25 (34.3%)	40 (56.3%)	
Tumor size					0.0095 **
	≥5 cm	96	56 (76.7%)	40 (56.3%)	
	<5 cm	48	17 (23.3%)	31 (43.7%)	
Stage					0.045 *
	I	79	35 (47.9%)	44 (61.9%)	
	II + III	65	26 (52.1%)	18 (38.1%)	
AFP					0.243
	≥25 ng/mL	72	33 (45.2%)	39 (54.9%)	
	<25 ng/mL	72	40 (54.8%)	32 (45.1%)	
Vascular invasion					0.128
	Yes	66	38 (52.1%)	28 (39.4%)	
	No	78	35 (47.9%)	43 (60.6%)	
Cirrhosis					0.517
	Positive	59	28 (38.4%)	31 (43.6%)	
	Negative	85	45 (61.6%)	40 (56.4%)	
Pathological stage					
	I	74	33 (45.2%)	41 (57.7%)	0.05 *
	II + III	70	40 (54.8%)	30 (42.3%)	

* *p* < 0.05; ** *p* < 0.01.

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
