# Peer review of "A Novel Long Non-Coding RNA-01488 Suppressed Metastasis and Tumorigenesis by Inducing miRNAs That Reduce Vimentin Expression and Ubiquitination of Cyclin E"

_cells, 2020, doi:10.3390/cells9061504_

Round 1
Reviewer 1 Report
Summary :
I reviewed with interest the manuscript cells-819997 by Lin and colleagues. In this study, the authors characterized the role of long non coding RNA LINC01488 in hepatocellular carcinoma (HCC). They report that LINC01488 downregulation in human HCC is associated with a better prognosis. Gain/loss of function experiments demonstrated that LINC01488 suppresses cell proliferation, migration and invasion in vitro. Mechanistically, LINC01488 modulates miR-124-3p/miR-138-5p to suppress metastasis through targeting EMT marker Vimentin. LINC01488 also induces ubiquitination and degradation of cyclin E. In vivo experiments in mice show that LINC01488 suppresses metastasis and tumorigenesis. Overall, the study is well conducted and includes gain/loss of function approaches, both in vitro and in vivo. Major issue of this study is the use of SK-Hep1 as a HCC cell line model.
Major points :
- In the introduction section, reference on the role of LINC01488 as a tumor suppressor should be clarified.
- SK-Hep1 is not a HCC cell line. Its use as HCC model has been refrained (PMID: 28807831). Key results should be validated in another well-accepted HCC cell line (e.g. Huh7, SNU-449).
- Gain of function of endogenous LINC01488 using CRISPRa is appreciated. The sequence of sgRNA should be provided.
- The first part of the result section on lincRNA profiling using only 2 HCC samples should be clarified. How statistical analysis was performed? The data should be shown being a starting point of the study.
- Figure 1B-D. Rather than using T/N ratio, the authors should analyze directly the expression in HCC tumors and perform the statistical analysis. The prognosis value of LINC01488 expression in HCC should be validated in an external validating dataset, e.g. by using already published datasets, including data from TCGA. It would be important to determine whether LINC01488 expression is an independent prognosis factor for survival.
- In Figure 2, it is not clear whether proliferation was blocked prior performing migration and invasion assays. This point is important given that LINC01488 impacts cell proliferation (Figure 3). The impact of LINC01488 on HCC cell proliferation should be presented first.
- 4B,D: WB should be quantified and statistical analysis performed. Immunofluorescence images (Supplementary figures) are not fully convincing. Figure S1A detected VIM in SK-Hep1 (control condition) and less in Hep3B. However, WB in Figure S2A shows opposite results at basal level (control condition). This point should be clarified. WB in Figure S2A need to be quantified and statistical analysis performed.
- 6, WB need to be quantified and statistical analysis performed.
Minor points :
Correct labelling of Fig. S4 versus S6 in the method section
The quality of figures should be enhanced
Line 305 should be rephrased
Line 310: LINC01488 / not linRNA-01488 for homogeneity
Reviewer 2 Report
The authors have examined a novel role of the long noncoding RNA (lncRNA) 01488 in metastasis and tumorigenesis of hepatocellular carcinoma (HCC). The manuscript describes how linc01488 suppressed metastasis by inducing miR-124-3p/miR-138-5p. In addition, the authors show that Linc-01488 also participates in cell cycle regulation. Here Linc-01488 was shown to bind to Cyclin E and trigger its ubiquitination.
The experiment was well designed; however I have a concern about the fundamental molecular investigation.
Major comments.
1.Linc01488 was reported as a noncoding RNA CUPID2 in breast cancer and modulated DNA damage response, but not cell proliferation (Betts et al., 2017). The authors described that “Long intergenic nonprotein-coding RNA 1488 (LINC01488) has been identified as a cyclin D long promoter-associated ncRNA located within the promoter region of cyclin D, which is located at chromosome 11q13.3 with the length of 3425 bp” (lines 54-56). There are several cyclin D promoter-associated lncRNAs (ncRNA CCND1, Wand et al., 2008), but Linc01488 is not one of them. It is transcribed near the distal enhancer of the CCND1 gene, located approx. 155kb upstream of the CCND1 locus. It looks like it is expressed using its own promoter. Please correct and discuss this point.
- The authors show a significant increase of mature miR-124-3p and miR-138-5p in Linc01488 overexpressing cells (Figure 5A). It is not clear to me how Linc01488 induces expression of miR-124-3p and 138-5p. The authors should at least examine whether Linc01488 undergo an increase in transcription or pre-microRNA processing?
- The authors show that Linc01488 also participates in cell cycle regulation by inducing ubiquitination of Cyclin E (Figure 6). Since miR-138-5p and miR-124-3p were reported to play a role as tumor suppressor (Wang et al., 2016; Yeh et al., 2019), the authors should examine whether the growth arrest phenotype observed in Linc01488 depleted cells is partially mediated by reduction of microRNAs.
- Does Linc01488 compete with CDK2 to bind to Cyclin E?
- To the best of my knowledge, SK-HEP-1 (ATCC No. HTB-52) is hepatic adenocarcinoma not hepatocellular carcinoma.
- Lines 56-58: “LINC01488 specifically binds an RNA-binding protein, Translocated in Liposarcoma (TLS), and exerts transcriptional repression through histone acetyltransferase (HAT) inhibitory activity”. It has been shown that ncRNACCND1 binds to TLS and suppresses CCND1 gene transcription (Wand et al., 2008). Since Linc01488 is not o ncRNACCND1, the authors should show the TLS-Linc01488 interaction.
Round 2
Reviewer 1 Report
I reviewed with interest the revised version of manuscript cells-819997-R1 by Lin and colleagues about the role of long non coding RNA LINC01488 in hepatocellular carcinoma (HCC). In this revised version, most of the comments raised by the reviewer have been addressed, including major and minor issues. However, a major issue of this study is still the use of SK-Hep1 as a HCC cell line model. This point needs to be corrected. An important note of caution has been already published on this topic to refrain the use of this cell line as a HCC model (PMID: 28807831). I suggest to the authors to clearly state on the origin of SK-Hep1 cells in the method section and to avoid associating SK-Hep1 with a HCC cell line model.
Reviewer 2 Report
6 Response to review 2: The authors wrote
“In the future, we will study LINC01488 whether located in CCND1 promoter region and bind withTLS by RIP assay.”
However, in the introduction lines 54-56: “Long intergenic nonprotein-coding RNA 1488 (LINC01488) has been identified as a cyclin D long promoter-associated ncRNA located within the promoter region of cyclin D, which is located at chromosome 11q13.3 with the length of 3425 bp.” This sentence is misleading. At least it should be toned down.
